# Early Detection of Sexual Predators with Federated Learning

**Khaoula Chehbouni**
Mila, HEC Montreal
khaoula.chehbouni@hec.ca

**Gilles Caporossi**
HEC Montreal
gilles.caporossi@hec.ca

**Reihaneh Rabbany**
Mila, McGill University
rrabba@cs.mcgill.ca

**Martine De Cock**
University of Washington Tacoma
mdecock@uw.edu

**Golnoosh Farnadi**
Mila, HEC Montreal
golnoosh.farnadi@hec.ca

## Abstract

The rise in screen time and the isolation brought by the different containment measures implemented during the COVID-19 pandemic have led to an alarming increase in cases of online grooming. Online grooming is defined as all the strategies used by predators to lure children into sexual exploitation. Previous attempts made in industry and academia on the detection of grooming rely on accessing and monitoring users' private conversations through the training of a model centrally or by sending personal conversations to a global server. We introduce a first, privacy-preserving, cross-device, federated learning framework for the early detection of sexual predators, which aims to ensure a safe online environment for children while respecting their privacy.

## 1  Introduction

The unprecedented rise in screen time and isolation brought about by the COVID-19 pandemic has left children more vulnerable than ever to online sexual exploitation. In 2021 alone, 85 million pictures and videos of child sexual abuse have been reported worldwide [13]. In May 2022, to fight against these growing numbers, the European Commission proposed a new regulation to compel chat apps to scan private user messages for child abuse and exploitation [13]. This new regulation was strongly condemned by privacy experts, who believed that implementing such mechanisms and breaking end-to-end encryption of users' messages could lead to mass surveillance [31].

Previous work on the identification of sexual predators has shown that the sexual predators' discourse contains specific indicators that can be leveraged for the detection of online grooming [26, 20, 25]. Some researchers focused on finding these linguistic cues by extracting lexical, syntactical, and behavioral features from chat messages [16, 22]. Others have used deep learning techniques to learn useful representations from text [33, 24]. Only a few treated the grooming detection problem as an early risk detection task [19, 32], i.e. recognizing grooming while it is happening and intervention is possible, as opposed to detection afterward. Furthermore, none of the proposed solutions were concerned with ensuring the privacy of the training examples. This represents a major limitation for the applicability of these models in a real-life setting, which is the main focus of this paper.

We present a novel privacy-preserving decentralized approach to train a context-aware language model [8] for the early detection of sexual predators in ongoing conversations. To do this, we leverage federated learning (FL) [23], an alternative to centralized machine learning (ML) that relies on a global server orchestrating the training of different entities without sharing any raw data, enhanced

Workshop on Federated Learning: Recent Advances and New Challenges, in Conjunction with NeurIPS 2022 (FL-NeurIPS'22). This workshop does not have official proceedings and this paper is non-archival.

with differential privacy (DP) [9] to provide formal privacy guarantees. Our key contributions are: (1) a practical, cross-device, privacy-preserving FL framework for the early detection of sexual predators in ongoing conversations; (2) an end-to-end implementation of our framework with an extensive evaluation on a real-world dataset.

The remainder of this paper is organized as follows: in Section 2 we present the existing related work. Section 3 introduces the preliminaries of our framework, which is then introduced in Section 4. In Section 5 we discuss its implementation details and evaluate it on a real-world dataset. Finally, we conclude with the ethical considerations surrounding such an application in Section 6 and discuss possible future works in Section 7.

## 2   Related Work

In this section, we review the most relevant works to our proposed approach in three main categories. First, we look at what has been done in the literature for the detection of sexual predators, then we introduce related work on the early detection of sexual predators before presenting existing work on decentralized text classification.

**Detection of sexual predators.** A competition organized at PAN-12 attracted attention to the task of identifying sexual predators with the creation of a new annotated dataset for the detection of grooming in messages [16]. Two problems were to be solved: (1) identify the predators among all the users and (2) identify the grooming messages. The winners of the first problem [30] used neural networks and SVMs to identify suspicious conversations on a pre-filtered version of the PAN-12 dataset, whereas the winners of the second problem [27] treated texts as sequences of symbols and used kernel-based learning methods to classify the grooming messages. Recent work mainly adopted deep learning techniques to solve the task [33, 24]. But all these approaches treated the problem from a forensic perspective rather than for prevention.

**Early risk detection.** To block harm from occurring, grooming should be detected before a victim is lured. Escalante et al. [11] made the first attempt at the early detection of sexual predators by adapting a naive Bayes classifier for grooming prediction with partial information. The authors evaluated the performance of their model with different percentages of words from the test set in a chunk-by-chunk evaluation framework that was later extended using profile-based representation [12, 19]. More recently, Vogt et al. [32] formally defined the task of early detection of sexual predators (eSPD), moving away from existing work to propose a sliding window evaluation, and creating a new dataset that is better suited for the task. We build on top of this work and use their proposed evaluation framework and dataset.

**Federated learning for text classification.** The approaches above assume training and deployment of models for grooming detection without concerns for privacy, i.e. while fully disclosing the users' personal messages to a central server for model training. FL, a method for training models in a decentralized fashion at the clients' end, and intermittently aggregating them via a central server, has been proposed as an alternative for natural language processing and text classification tasks (see e.g. [14, 15]). While privacy is preserved to some extent in FL because no raw data is disclosed, information about the clients' training data may leak from the gradients or model parameters sent to the central server [5, 6]. This information leakage can be mitigated by combining FL with another privacy-enhancing technology such as differential privacy (DP), e.g. by training models with differentially private gradient descent (DP-SGD) [1]. Basu et al. [3] have for instance recently applied FL and DP-SGD for financial text classification. To the best of our knowledge, privacy-preserving early detection of abusive content in a decentralized manner by leveraging both FL and DP-SGD, as we propose in this paper, has not been investigated in the literature.

## 3   Preliminaries

In this section, we review several key topics upon which our proposed privacy-preserving early detection of sexual predator framework relies. In our work, we leverage federated learning and differential privacy to protect the privacy of users, hence we first introduce DP and then provide a brief overview of the DP-SGD algorithm.

Whilst FL protects the privacy of the clients by not requiring any raw data to be disclosed, FL in itself does not offer formal privacy guarantees, and the resulting model can leak information about the training data [6]. To mitigate such information leakage, FL can be combined with DP [9] to provide plausible deniability regarding an instance being in a dataset, i.e. offering protection against membership inference attacks.

Formally, DP revolves around the idea of a randomized algorithm – such as an algorithm to train ML models – producing very similar outputs for adjacent inputs. In the context of this paper, two datasets $d$ and $d'$ are considered adjacent if they differ in one record (one labeled instance). A randomized algorithm $M : D \mapsto R$ with domain $D$ and range $R$ is said to be $(\epsilon, \delta)$-differentially private if for any adjacent datasets $d$ and $d'$ and for all subsets of outputs $S \subseteq R$ we have $Pr[M(d) \in S] \leq e^\epsilon Pr[M(d') \in S] + \delta$, where $\epsilon$ is the metric of privacy loss (privacy budget) whereas $\delta$ is the probability of data being accidentally leaked. The smaller these values, the stronger the privacy guarantees.

An $(\epsilon, \delta)$-DP randomized algorithm $\mathcal{M}$ is commonly created out of an algorithm $\mathcal{M}^*$ by adding noise that is proportional to the sensitivity of $\mathcal{M}^*$, in which the sensitivity measures the maximum impact a change in the underlying dataset can have on the output of $\mathcal{M}^*$. This technique is used in the differentially private stochastic gradient descent (DP-SGD) algorithm which aims at controlling the influence the training data has on the final model by making the minibatch stochastic optimization process differentially private through clipping and adding noise to the gradients [1]. At the end of the training, the overall privacy cost of the mechanism $(\epsilon, \delta)$ can be computed from the accumulated costs across all training iterations. Often, a target $\epsilon$ is defined in advance whereas $\delta$ should be smaller than the inverse of the size of the training data. We refer to Abadi et al. [1] for details.

# 4   Methodology

While protecting children from cybercrime is important, the main challenge is the balance between safety and users' privacy. In this section, we introduce the two steps of our privacy-preserving framework for the identification of sexual predators. Our proposed framework consists of, first, training a model in a federated manner on the local personal conversation of users with local DP (training phase), and then evaluating its performance for the early detection task on the test set (inference phase).

## 4.1   Training Phase: eSPD via Federated Learning

We introduce a cross-device federated architecture for the early detection of online grooming: our model is intended to be deployed on each user's cellular device and trained locally on their local data without the need for monitoring them.

Our framework addresses multiple task-specific challenges: (1) training with imbalanced data, (2) training with non-independent and identically distributed (non-IID) data and (3) ensuring that users' personal data are protected during training.

**(1) Dealing with imbalanced data.** To deal with the problem of imbalanced data – namely very few positive instances – that often comes with early risk detection problems, we implement Errecalde et al. [10]'s oversampling technique. They considered that the minority class is formed not only by the complete conversation but also by portions of the full conversation at different time steps. Therefore, to account for the sequential nature of the eSPD problem we enrich our dataset with chunks of conversations from the minority class, in our case, the conversations with a predator. By giving our system more training examples of the beginning of a conversation with a predator, we are able to gain detection speed. Furthermore, in a federated setting, having more data gives more weight to a client during aggregation when we use the FedAvg algorithm, which helps alleviate the imbalance problem.

**(2) Training with non-IID data.** One of the major challenges of FL is dealing with non-IID data since each client's local data distribution is not representative of the population [35]. This statistical challenge is even more prevalent in the context of online grooming since most users are less likely to interact with sexual predators. Thus, the detection of online grooming in a federated setting can be viewed as an extreme case of non-IID data where most users will only have access to one label

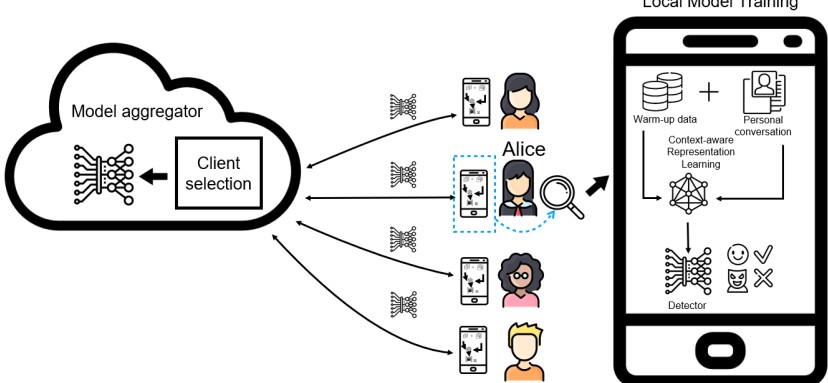

Figure 1: Early Detection of Sexual Predators: Training Phase

for training. Indeed, only the victims of online grooming will have access to both grooming and non-grooming conversations.

We use Zhao et al. [34]'s data-sharing strategy during training in which a small portion of *warm-up data* is distributed to each device in addition to the initial model. The *warm-up data*, which contains public examples from both classes and is balanced, can be seen as a starting point for training, and helps alleviate the statistical challenge. In their paper, Zhao et al. [34] also suggested sharing a *warm-up model* with each client: a model trained centrally on the warm-up data. We experimented with this strategy but realized that each client did not have enough data and with this strategy the *warm-up model* may not learn from users' raw data. Instead, we decided to only share a small fraction of the *warm-up data* with each user during the training phase.

**(3) Protecting users' privacy.** Although each client's local data does not leave their device during federated training, it has been shown that it is possible to reconstruct a client's private data using its shared updates [17], hence a federated architecture by itself does not guarantee privacy. We therefore train each client's model using DP-SGD (see Section 3), to mitigate leakage of personal information to the server. By clipping the gradient norm of outliers and randomly adding noise during training, we ensure that our model does not memorize any particular information about a single training data point.

Figure 1 illustrates the training phase of our framework. A global server selects clients to participate and distributes a model to them; the clients will then further train the model in a privacy-preserving manner on their mobile devices using their own personal data as well as a portion of warm-up data, as we can see in Alice's cellular device.

### 4.2 Inference Phase: Early Detection of Sexual Predators

Our work is an extension of the framework proposed by Vogt et al. [32] for eSPD, i.e. the early risk detection problem [21] of sequentially classifying a conversation and detecting early signs of online grooming as soon as possible.

Vogt et al. [32]'s approach for the inference phase of an eSPD system relies on the use of a sliding window for the sequential classification of a conversation. Here, a conversation consists of a sequence of messages $t_1, t_2, \ldots$

For a window of length $l$, at step $s$ the classifier labels the sequence $t_s, t_{s+1}, \ldots, t_{l-1}$, at step $s + 1$ the classifier labels the sequence $t_{s+1}, t_{s+2}, \ldots, t_l$ etc.

After every window prediction, the system decides whether to raise a warning or not based on the inferred labels of the last 10 window predictions. If a pre-defined threshold – called skepticism level – is reached, a warning is raised and the whole conversation is classified as a grooming conversation. A conversation is only classified as a non-grooming conversation if it is finite and no warning has been

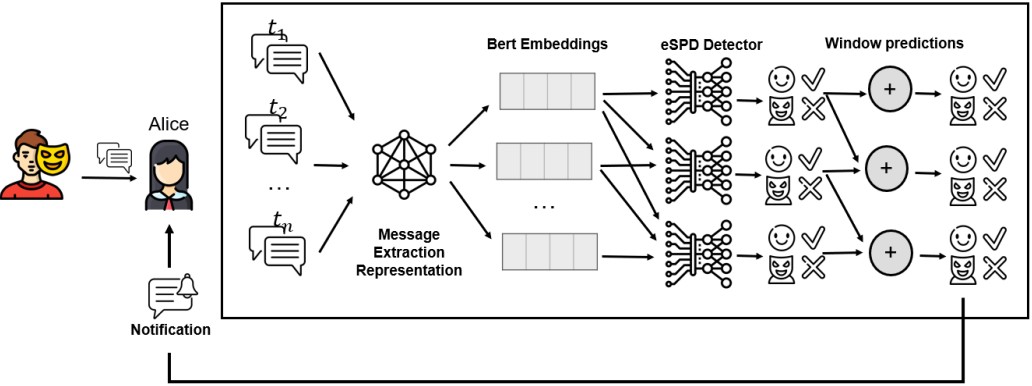

Figure 2: Early Detection of Sexual Predators: Inference Phase

raised. Indeed, an eSPD system never classifies a conversation as non-grooming if there are messages left, or if it is still ongoing.

In Figure 2, we can see how the different messages received by Alice are analyzed by first being turned into word embeddings and then passed to a classifier given a sliding window for classification. Note that the final prediction is determined based on the previous sequence of predictions and that a warning notification is triggered only when multiple messages are sequentially classified as being grooming messages.

We can envision a system where users will be able to report their own suspicious conversations to the messaging platforms, and will receive a notification if a warning is raised (see Figure 3).

## 5 Evaluation

In this section, we show the effectiveness of our proposed approach for the early detection task by performing an empirical evaluation. All our experiments were performed on the *PANC dataset*.

### 5.1 Data

The **PANC dataset** was introduced by Vogt et al. [32] as a better alternative for the eSPD task. It was created by merging the "negative" (non-grooming) conversations from the PAN 12 competition [16], sampled from IRC logs and the Omegle forum,[1] and the "positive" (grooming) conversations from the ChatCoder2 dataset [22]: 497 complete conversations extracted from the Perverted Justice (PJ) website[2]. They filtered the full grooming conversations and split them into segments to make them comparable to the non-predatory examples and create a corpus better suited for the task of early detection. Despite its numerous limitations, such as the lack of full negative conversations and small differences in formatting between the two classes, we found that the PANC dataset is the most appropriate available data for our task.

The PANC dataset (see Table 1) was split into a training set (60%) and a test set (40%). The training set consists of 1,753 positive segments (representing in total 298 full-length positive conversations and 9.06% of the training examples) and 17,598 negative segments, whereas the test set contains 10.84% examples of grooming.

In Figure 3, we present a visualization of a synthetic setup based on our proposed framework using a predatory conversation from the PANC dataset. It can take weeks or even months before a warning notification is triggered when a child is being lured by an abuser. Our goal is to minimize the harm by detecting the abuse early and sending a notification to the user. It is up to the user to decide whether to continue the conversation or report the predator. Note that in our framework, both training and

---

[1]https://www.omegle.com
[2]http://www.perverted-justice.com/

Table 1: Statistics about the PANC dataset [32]

| Label | Number of segments | | Words/segment | | Messages/segment | |
|---|---|---|---|---|---|---|
| | train | test | train | test | train | test |
| 0 | $17,598$ $(91\%)$ | $11,733$ $(89\%)$ | $173$ $(\pm1,385)$ | $184$ $(\pm1529)$ | $36$ $(\pm25)$ | $36$ $(\pm26)$ |
| 1 | $1,753$ $(9\%)$ | $1,426$ $(11\%)$ | $289$ $(\pm218)$ | $292$ $(\pm222)$ | $64$ $(\pm43)$ | $65$ $(\pm43)$ |

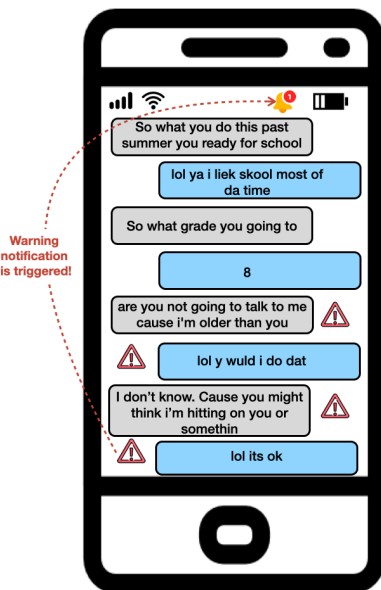

Figure 3: Visualization of eSPD in which the risk is detected, a warning is raised after passing a threshold, and the user is notified as early as possible.

inference phases are happening locally and users' personal conversations are never shared with a third party. Moreover, the global aggregated model from the server can further be tuned and personalized based on users' local data.

## 5.2 Evaluation Metrics

In addition to the established metrics of precision, recall, and F1 score, we use the latency-weighted F1 score introduced by Sadeque et al. [28] for the early risk detection task. The F-latency metric measures the trade-off between the speed of detection (i.e. how early in a converation grooming is detected) and the accuracy of the warning by applying a penalty that increases with the warning latency. A higher F-latency score means a better-performing eSPD system. The warning latency is defined as the number of messages exchanged before a warning is raised [32]. The penalty can be computed for each warning latency $l \geq 1$ as follows:

$$\text{penalty}(l) = -1 + \frac{2}{1 + e^{(-p \cdot (l-1))}}$$

where $p$ defines how quickly the penalty should increase. As suggested by Sadeque et al. [28], $p$ should be set such that the latency penalty is $50\%$ at the median number of messages of a user.

The "speed" of an eSPD system over a test set of grooming conversations is defined as speed $= 1 - \text{median}\{\text{penalty}(l) \mid l \in \text{latencies}\}$ where "latencies" corresponds to the list of warning latencies produced by the system for all grooming conversations for which a warning is raised.

We can then formally define F-latency as: F-latency $= $ F1 $\cdot$ speed.

While F1 is computed across the entire test set of positive and negative messages, penalty and speed are computed for the positive conversations only. This is common practice in the literature as the delay needed to detect true positives is a key component of the early risk detection task [21, 28].

## 5.3 Experimental Set-Up

**Data manipulation.** As explained in Section 4, we leverage the oversampling technique proposed by Errecalde et al. [10] to our training data to improve the speed of our system's detection. As such, we add four additional segments to each grooming conversation in our training set: the first 10% characters of the full conversation, then 20%, 30%, and 40% of the full conversation. We selected the number of augmented data portions with the help of hyperparameter tuning.

Furthermore, to implement the data-sharing strategy, we split the augmented PANC training set into three: 10% of the dataset is randomly selected to create the warm-up data, and the rest is split between a training set (81%) and a validation set (9%). Finally, since neither the test set nor real-life data will be augmented, we remove the additional chunks of data from the validation set.

To ensure that no bias came from the warm-up split, we repeated the process three times and tested our model with every split. We have also experimented with different sizes of warm-up data (1% and 5%) and concluded that a 10% split was better suited for the task.

**Federated set-up.** In our cross-device federated framework, we create each client by randomly selecting one user from the training set. In our dataset, each user corresponds to a unique conversation, either predatory or non-predatory. And as seen in Subsection 5.1, whereas each "positive" user has multiple segments of data, each "negative" user only has one segment of data.

Therefore, to compensate for the lack of non-predatory examples, if the user selected is a "negative" user, we then select 10 additional "negative" users and combine their data. Furthermore, at initialization, each client receives a random, balanced portion of the warm-up data: 10 segments with a "negative" label and 10 segments with a "positive" one to complement their own data.

**Choice of the classifier.** Although fine-tuning BERT has been shown to give better results for the early detection task [32], we use the pre-trained feature-based approach with logistic regression (LR) since it is far less computationally expensive and better suited for scaling federated training to a large number of clients.

In our framework, each user uses the $\text{BERT}_{\text{BASE}}$ model to create a context-aware representation of their personal conversation by extracting fixed features from the pre-trained model. The [CLS] representation of the last layer is then used as an input for LR with a binary cross entropy loss function. For each user's segment, we, therefore, obtain a 768 length vector.

**Implementation.** We use Flower [4], an FL framework that facilitates large-scale experiments through its simulation tools, to implement our setup and collaboratively train a logistic regression model with 10,000 clients for 100 rounds. At each round of training, we select $10\%$ of the clients randomly to participate in the training, and the parameters are aggregated with the FedAvg algorithm [23].

The optimal number of rounds was determined by following the evolution of the validation loss of different models during training.

**Training with DP-SGD.** Every client selected for the training process will train its data with logistic regression with differentially private stochastic gradient descent. A random grid search was conducted to test for different hyperparameters: notably, the selected range for the gradient clipping level is $(0.5, 1, 2, 5, 7)$, and we tried $(0.01, 0.05, 0.001, 0.0001)$ for the learning rate, $(8, 16, 32, 100)$ for the batch size, and $(5, 10, 15, 20, 100)$ for the number of local epochs of training.

All the models were evaluated using a 50-message sliding window and a skepticism level of 5, i.e. 5 of the last 10 predictions had to be positive before a warning was raised. Finally, Appendix A presents the resources used for training our models. Our eSPD implementation can be found at https://github.com/khaoulachehbouni/fl-espd.

## 5.4 Empirical Results

We investigate three research questions in our experiments:

Table 2: Evaluation results of the early online grooming detection task

| Model | F1 | Recall | Precision | Speed | F-latency | FPR |
|---|---|---|---|---|---|---|
| Baseline | 0.50 | **0.98** | 0.33 | 0.96 | 0.48 | 0.24 |
| Centralized | 0.75 | 0.95 | 0.62 | **0.83** | 0.63 | 0.07 |
| Cross-Device FL | **0.82** | 0.85 | **0.79** | 0.79 | **0.64** | **0.03** |
| Cross-Device FL+DP-SGD ($\epsilon = 1$) | 0.76 | 0.86 | 0.68 | 0.81 | 0.61 | 0.10 |

**RQ1: How is the utility of the eSPD system affected by the FL framework?**

To address the first research question, we compare the utility of our cross-device approach with two baselines: (1) *Baseline (warm-up data)*: A logistic regression model trained centrally on the warm-up data only, to ensure that our framework is not too biased by the warm-up data distributed to each client; and (2) *Centralized LR*: A logistic regression model trained centrally on the training data and the warm-up data.

Both centralized models used five-fold cross-validation for hyperparameter tuning whereas the best hyperparameters for the federated models have been chosen using a random search.

In Table 2 we can see that the federated frameworks show competitive results for the early detection task. Indeed, the cross-device FL model has the higher F-latency score, and the loss of utility that comes with making our model differentially private is moderate. We believe that this good performance can be attributed to the fact that cross-device training gives more importance to the minority class than centralized training. Indeed, the FedAvg algorithm takes into consideration the amount of data held by each client to aggregate the models, and in our case, the "positive" users train with more examples. The warm-up data also alleviates the imbalance problem by giving each user enough examples of both class.

In addition to showing slightly better results for the early detection task (with a 64% F-latency score), the cross-device framework also has the lowest false positive rate (3%).

Furthermore, we see that the speed of the baseline model is very high but it also comes with a higher false positive rate (FPR): detection speed always comes at the cost of precision. The baseline model also has a very low F-latency: our model is therefore not biased by the data sharing strategy and it is indeed learning from each client's personal data.

**RQ2: How to reduce the harm of false positives in eSPD?**

In eSPD, the emphasis is often put on the detection of predators since missing one could cause a lot of harm. Indeed, the F-Latency metric depends on both the F1-score and the speed. And while the F1-score takes into consideration both recall and precision, the speed does not penalize for precision: a model that predicts every conversation as being predatory will have a very high speed. Therefore, we propose an approach to adjust our model to consider the cost of falsely accusing someone as predator.

For this purpose, for each of our models, we identify the classification threshold that is needed to achieve a 1% false positive rate (FPR) when evaluated on the test set. Using this new threshold, we re-evaluate our models. Table 3 shows that varying the threshold comes with a loss in speed, which is to be expected since higher prediction scores are now needed to classify a window as a grooming conversation. Furthermore, the results for the baseline model are not presented because the smaller FPR attained for this model with a 0.99 classification threshold is 9%, showing that it was falsely classifying non-predatory conversations as predatory.

Finally, we notice a decrease in F-latency for all the models, a necessary trade-off to achieve better precision.

Figure 4 shows the distribution of the warning latencies after we change the classification thresholds of each model to attain a target low false positive rate (FPR=1%). We can see that a larger number of messages is needed in average to attain better precision. In early risk detection, a trade-off is always necessary between the speed of detection and the precision of a warning.

**RQ3: How does differential privacy impact the eSPD system?**

To evaluate the cost of privacy on eSPD systems, we experiment with adding various amounts of noise $\epsilon$ to the training process.

Table 3: Evaluation results for a 1% FPR

| Model | F1 | Recall | Precision | Speed | F-latency |
|---|---|---|---|---|---|
| Baseline | – | – | – | – | – |
| Centralized | 0.85 | 0.83 | 0.88 | 0.69 | 0.59 |
| Cross-Device FL | 0.83 | 0.78 | **0.89** | **0.73** | **0.61** |
| Cross-Device FL+DP-SGD ($\epsilon = 1$) | 0.78 | 0.70 | 0.88 | 0.72 | 0.57 |

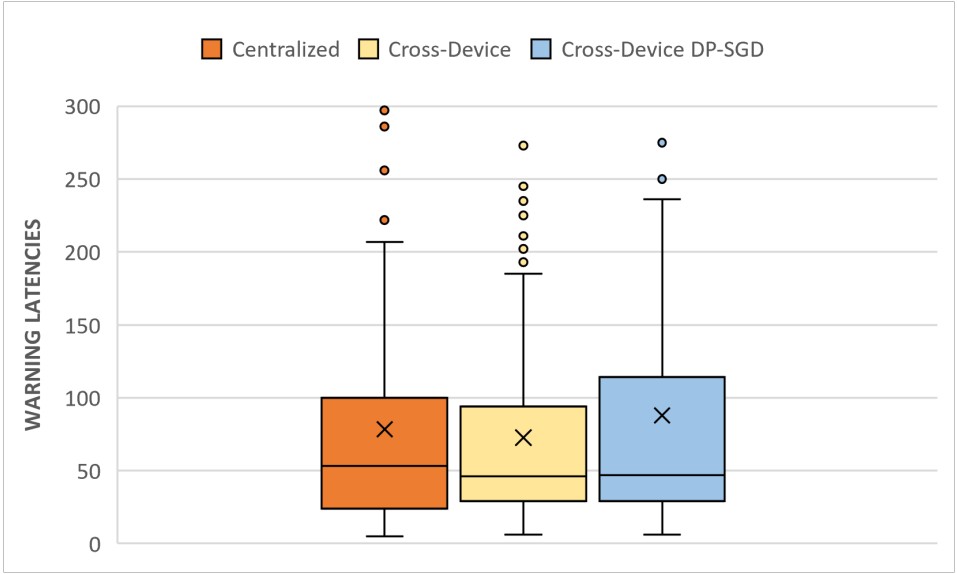

Figure 4: Warning latencies for a skepticism level of 5 with the classification threshold needed to achieve a 1% FPR

We observe that the less performing model is the one with the highest privacy constraints: with an $\epsilon$ of 0.50, we notice a drop of 8% of the F-latency score for the most private model as seen in Figure 6. However, we notice that there is no loss in utility when $\epsilon$ is greater than 10. Furthermore, as we can see in Figure 5, the precision graph has a steeper slope and therefore seems to be more impacted by the differentially-private training.

Indeed, it has been shown that DP-SGD does not affect the performance of a model equally and that minority classes may be more affected by the training process [2]. In our case, making our model more private may result in a decrease in its ability to detect predators adequately.

## 6 Limitations and Ethical Considerations

In this section, we explore the limitations of our proposed approach and ethical considerations relating to the implementation of such a tool in a real-life setting.

Beyond the privacy issues, a main challenge in addressing the sexual predators' identification task through machine learning comes from the lack of publicly available labeled and realistic datasets. The different datasets used in the literature all take their grooming examples from the PJ website, which are examples of conversations between predators and adults posing as children to catch them. Such chats have been shown to differ from real-life conversations and lack certain aspects of grooming like overt persuasion and sexual extortion [29]. Indeed, volunteers are often actively trying to get the offenders to be sexually explicit and to arrange an encounter, which is not the case in real-life settings. Furthermore, the non-grooming examples often come from forums and chatrooms where strangers can interact or engage in cyber-sex. Lack of negative examples of trusting and intimate relationships between family members, friends, or partners is an issue of the current datasets which are essential components for a realistic eSPD task.

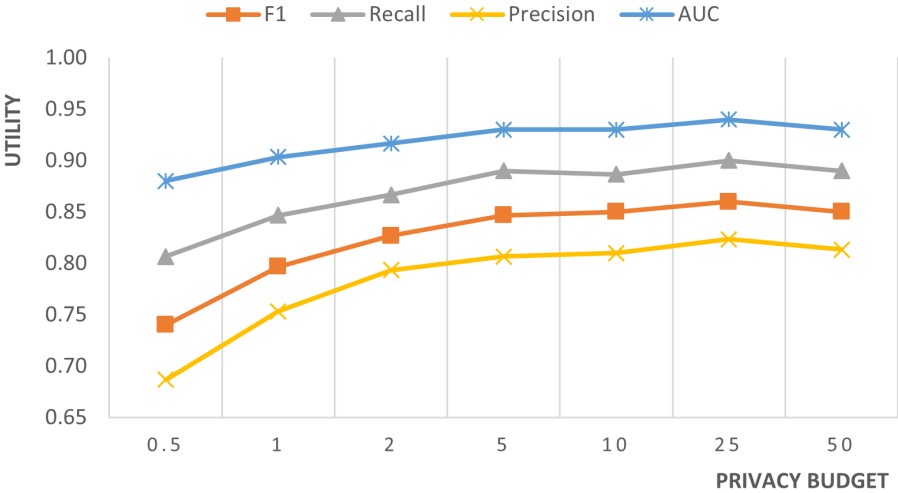

Figure 5: Impact of the privacy budget on the utility of a cross-device federated model. All the models were evaluated on the full test set.

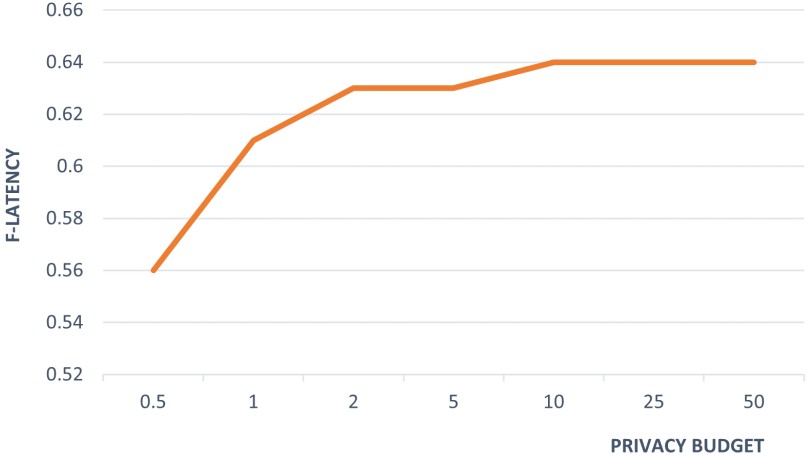

Figure 6: Impact of the privacy budget on the early detection performance of a cross-device federated model. All the models were evaluated on the full test set.

We hope that the federated architecture we propose in this paper, will give access to a larger range of training examples. Indeed, since each user will be given the option to report abusive content, the conversations flagged as alleged grooming will then be added to the pool of training examples, thus alleviating the lack of realistic and available labeled datasets. Such a system will allow the training examples to be updated regularly and will consider the growing speed at which language, especially internet slang, evolves.

However, we can imagine that even with such a framework, the labeling will still be an issue since it will rely on users self-reporting cases of grooming. We could think of a preliminary training phase with real data of convicted predators before deploying a pre-trained model to evaluate each user's personal conversation and send a notification where a warning is raised by the eSPD system. Such a model will also alleviate the privacy cost since the first training phase will happen on publicly available data. In this setting, the user will be able to give feedback on the model's prediction. But such a set-up is certainly not ideal, since actual victims of online grooming often trust their abuser and may not realize that they are being manipulated. Notifying a third party, such as a legal guardian or a social worker tasked with monitoring the flagged content, may increase the chances of a case of grooming being reported but will undoubtedly infringe on the privacy of the victim.

Involving law enforcement could also have disastrous consequences. As we have mentioned in subsection 5.4, the resulting model could be biased towards certain populations like sex workers, people from the LGBTQI+ community, or people prone to online dating. Evaluating and selecting the best model based on a classification threshold that guarantees a $1\%$ false positive rate can be a first step towards ensuring that the eSPD system does not falsely incriminate. Furthermore, pre-trained language models used to extract a context-aware representation of personal conversations, like BERT, have been shown to reproduce racial and gender biases [18]. Using such models as a basis for identifying potential suspects to be prosecuted could lead to unanticipated outcomes. Such a system should therefore never be used directly by law enforcement agencies at the risk of exacerbating existing social inequalities and persecuting innocents.

Finally, the literature and datasets used for our experiments concern male predators, both heterosexual and homosexual, that do not know their victims. The lack of data available about female abusers does not allow us to assume that our model is applicable to the detection of female predators.

## 7 Conclusion and Future Directions

In the wake of the new European Commission's regulation [13], social media companies will be expected to take action to ensure that their underage users are safe from sexual exploitation when using their platforms. Doing so would entail breaking end-to-end encryption and monitoring users' content, which can easily lead to human rights infringements as we have seen recently with the case of the teenager charged for abortion in Nebraska after Meta turned over her personal chat messages to the police [7]. Alternatives to existing privacy-invasive monitoring systems are therefore more pressing since the COVID-19 pandemic has increased the need for children's safety. In this paper, we presented a first-of-its-kind federated learning framework for the early detection of sexual predators and we showed that the utility of our system is comparable to the utility of a model trained in a centralized manner while fully protecting users' personal data rights.

We believe that protecting children from sexual exploitation should not come at the cost of privacy or additional abuse. Finally, it is also essential to consider the possible biases such a model could have and the high cost of falsely accusing someone as a predator since large pre-trained models come with racial and gender biases inherited during training [18]. Addressing these challenges remain as future direction of this work. Finally, we believe that our framework can be extended to any early risk detection problem: future work could explore the use of our framework for the detection of cyberbullying or depression.

## Acknowledgments

This project was partially funded by Mila AI for Humanity and Canada CIFAR AI chair. This research was enabled in part by support provided by Calcul Québec and Compute Canada. We would also like to thank Dr. April Edwards for sharing with us the ChatCoder 2 dataset and Mathias Vogt for taking the time to answer our questions regarding their early detection framework.

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

# A Appendix: Computational ressources

## A.1 Training phase

Table 4: Time and resources used to train the different models

| Model | Memory | CPUs | Time |
|---|---|---|---|
| Baseline | 16 GB | 4 | 1 min |
| Centralized | 16 GB | 4 | 1 min |
| Cross-Silo FL | 32 GB | 4 | 2 min |
| Cross-Device FL | 32 GB | 4 | 3 h |
| Cross-Device FL+DP-SGD ($\epsilon = 1$) | 32 GB | 8 | 3 h |

Table 4 presents the resources used to train the different models. Note that the embedding extraction with BERT was done separately and with an NVIDIA Tesla T4 GPU (it takes around 20 minutes).

Many of the experiments performed in this work were limited by scalability issues. For example, we could not use a GPU while running an experiment with more than 10,000 clients since, in the Flower simulation setting, resources must be shared between all participating devices, which is not the case in a real-life setting where each device brings its own resources.

## A.2 Inference phase

The early detection evaluation takes around 2 hours with an NVidia A100 GPU (with 40 GB of memory).

