# OpenReview forum: "Early Detection of Sexual Predators with Federated Learning"
_NeurIPS.cc/2022/Workshop/Federated_Learning — FL-NeurIPS 2022 Poster_

### Official Review · Reviewer_96px · 2022-10-03
**This paper claims to be the first of its kind on detecting online grooming, in order to further prevent sexual exploitation of children. The proposed framework is a collage of existing methods; yet, the topic is of great importance, making this a great contribution of the uses FL could have**

Overall: This paper considers the problem of grooming detection at its early stages, in order to achieve grooming detection and lead to on-time intervention. This deviates from most existing results that discuss the case of grooming detection, after it might have happened (based on more data, etc). The paper uses FL framework to achieve that: it presents a novel privacy preserving decentralized approach to train a context-aware language model for the early detection of sexual predators. This is achieved via federated learning settings and by implementing an end-to-end setting with real datasets.

Strengths:
+ The topic itself. It highlights the usage of FL beyond classical ML applications
+ The empirical evaluation: this paper (among with some other papers cited) could pave the way on this topic about baselines on this topic

Weaknesses:
- The approach is a combination of existing works, which does not add to the intellectual merit of this work.

Overall, I encourage such works that highlight the value of what we are studying, working on and "fighting" for. I suggest acceptance.

---

### Official Review · Reviewer_a3p9 · 2022-10-17
**Well written but both novelty and contribution are not enough**

This paper focused on the early detection of sexual predators through federated learning, considering imbalanced and non-iid data and differential privacy. The paper is well written and organized. However, there are some cons:

1. The novelty and originality of this paper are not enough. The three main challenges considered in this paper:  imbalanced data, non-iid data, and privacy preservation, have been extensively studied in literature, and the authors just adopted the existing solutions.

2. The technical contribution is not clear from either analytical or numerical perspective.

---

### Official Review · Reviewer_9i6H · 2022-10-17
**Unclear privacy modeling**

This paper purports to detect sexual predators online while still preserving privacy. This is done through federated learning with differential privacy.

While this is a laudable attempt and a good problem, it was not clear to me after reading the paper whether differential privacy is the right notion of privacy for this use-case. I would encourage the authors to discuss their choice of privacy modeling and provide more evidence via concrete examples that the right kind of privacy is being offered.

---

### Decision · Program_Chairs · 2022-10-20

Accept (Poster)